



# Influence of organic aerosol composition determined by offline
# FIGAERO-CIMS on particle absorptive properties in autumn
# Beijing
Jing Cai[1,2], Cheng Wu[3], Jiandong Wang[4], Wei Du[1,2], Feixue Zheng[1], Simo Hakala[1,2], Xiaolong Fan[1],
Biwu Chu[1,2,5], Lei Yao[2], Zemin Feng[1], Yongchun Liu[1], Yele Sun[6], Jun Zheng[7], Chao Yan[1,2], Federico
Bianchi[1,2], Markku Kulmala[1,2,8,9], Claudia Mohr[3*], Kaspar R. Daellenbach[1,2,10*]
[1] Aerosol and Haze Laboratory, Beijing Advanced Innovation Center for Soft Matter Science and Engineering, Beijing
University of Chemical Technology, Beijing 100029, China
[2] Institute for Atmospheric and Earth System Research, Faculty of Science, University of Helsinki, Helsinki 00014, Finland
[3] Department of Environmental Science, Stockholm University, Stockholm, 11418, Sweden
[4] School of Atmospheric Physics, Nanjing University of Information Science and Technology, Nanjing 210044, China
[5] State Key Joint Laboratory of Environment Simulation and Pollution Control, Research Center for Eco-Environmental
Sciences, Chinese Academy of Sciences, Beijing 100085, China
[6] State Key Laboratory of Atmospheric Boundary Layer Physics and Atmospheric Chemistry, Institute of Atmospheric
Physics, Chinese Academy of Sciences, Beijing 100029, China
[7] Jiangsu Key Laboratory of Atmospheric Environment Monitoring and Pollution Control, Nanjing University of Information
Science & Technology, Nanjing 210044, China
[8] Joint International Research Laboratory of Atmospheric and Earth System Sciences, School of Atmospheric Sciences,
Nanjing University, Nanjing, China
[9] Faculty of Geography, Lomonosov Moscow State University, Moscow, Russia
[10] Laboratory of Atmospheric Chemistry, Paul Scherrer Institute, Villigen, Switzerland.
*Correspondence to:* claudia.mohr@aces.su.se and kaspar.daellenbach@psi.ch




## Abstract:

Organic aerosol (OA) is a major component of fine particulate matter (PM) affecting air quality, human health, and the climate. The bsorptive and reflective behavior of OA components contributes to determining particle optical properties and thus their effects on the radiative budget of the troposphere. There is limited knowledge on the influence of the molecular composition of OA on particle optical properties in the polluted urban environment. In this study, we characterized the molecular composition of oxygenated OA collected on filter samples in autumn of 2018 in Beijing, China, with a filter inlet for gases and aerosols coupled to a high-resolution time-of-flight chemical ionization mass spectrometer (FIGAERO-CIMS). Three haze episodes occurred during our sampling period with daily maximum concentrations of OA of 50, 30, and 55 µg m$^{-3}$, respectively. We found that the signal intensities of dicarboxylic acids and sulfur-containing compounds increased during the two more intense haze episodes, while the relative contributions of wood-burning markers and other aromatic compounds were enhanced during the cleaner periods. We further assessed the optical properties of oxygenated OA components by combining the detailed chemical composition measurements with collocated particle light absorption measurements. We show that light-absorption enhancement ($E_{abs}$) of black carbon (BC) was mostly related to more oxygenated OA (e.g. dicarboxylic acids), likely formed in aqueous-phase reactions during the intense haze periods with higher relative humidity, and speculate that they might contribute to lensing effects. Aromatics and nitro-aromatics (e.g. nitrocatechol and its derivatives) were mostly related to a high light absorption coefficient ($b_{abs}$) consistent with light-absorbing (brown) carbon (BrC). Our results provide information on oxygenated OA components at the molecular level associated with BrC and BC particle light-absorption and can serve as a basis for further studies on the effects of anthropogenic OA on radiative forcing in the urban environment.

## 1. Introduction

Organic aerosol (OA) makes up a large fraction of submicron aerosol particles globally (Jimenez et al., 2009). As such, OA plays an essential role in numerous atmospheric processes such as photochemical oxidation, new particle formation and growth, and cloud formation, and influences atmospheric pollution and human health, as well as global radiative forcing (Jimenez et al., 2009;Riipinen et al., 2012;Lu et al., 2019;Lelieveld et al., 2015;Daellenbach et al., 2020). Secondary organic aerosol (SOA) or oxygenated organic aerosol OOA (a surrogate of SOA) comprises a large number of organic compounds, many of them unknown, formed via oxidation of gas-phase organic precursors (volatile organic compounds, VOCs). SOA accounts for a large fraction of the total OA burden in the atmosphere (Jimenez et al., 2009). Knowledge gaps remain regarding SOA sources and formation mechanisms, especially in polluted areas with strong anthropogenic emissions (Huang et al., 2014).

OA, is found to be an important source of brown carbon (BrC), as light-absorbing OA is denoted. OA can also act as an effective shell of internally mixed black carbon (BC) particles that focuses photons onto the BC core (named 'lensing effect' (Jacobson, 2001)), which leads to so-called light-absorption enhancement ($E_{abs}$) of BC particles (Xie et al., 2019a;Xie et al., 2019b;Zhang et al., 2018;Liu et al., 2015;Wang et al., 2018). For all these optical effects, the chemical composition of OA plays a role (Zhang et al., 2011;Fleming et al., 2020;Laskin et al., 2015); OA light absorption can therefore not be fully quantified based on bulk concentrations only. Certain OA compounds, e.g. nitrophenol derivatives and amorphous carbon spheres (i.e., tarballs), formed from anthropogenic precursors, were found to be important components of BrC (Cheng et al., 2016a;Mohr et al., 2013;Wang et al., 2019b) and to significantly enhance the light absorption properties of particles even when present in small amounts (Teich et al., 2017). In contrast, certain biogenic SOA compounds seem to be less light-absorbing (Zhang et al., 2011). Generally, OA with a higher degree of oxygenation leads to higher BC $E_{abs}$ than less oxygenated OA (Zhang et al., 2018). In fact, less oxygenated OA was estimated to have a negligible or even negative effect on $E_{abs}$ in a study conducted in Beijing, China (Xie et al., 2019a). To better understand the impact of OA composition on particle optical properties, and to estimate effects on radiative forcing on both regional and global scales, detailed OA chemical composition and BrC/BC optical measurements need to be combined.

OA components can be characterized at the molecular level using offline gas or liquid chromatography coupled to mass spectrometry (GC/MS or LC/MS), which allows identification and quantification of a limited number or groups of compounds, due to the lack of standards (Schauer et al., 2002;Guo et al., 2012). More recently established online mass spectrometer methods can provide detailed composition information for many OA compounds, albeit without structural information. For





example, Aerosol Mass Spectrometers (AMS) are widely used to yield insights into the chemical evolution of OA when
combined with factor analytical methods (Cai et al., 2015;Du et al., 2017;Hu et al., 2017;Sun et al., 2016;Jimenez et al., 2009).
Mass spectrometers employing chemical ionization coupled with different inlets such as the filter inlet for gases and aerosols
(FIGAERO) (Thornton et al., 2020) or the Chemical Analysis of Aerosol Online (CHARON) (Müller et al., 2017) allow for
SOA composition analysis in both the gas and particle phase at the molecular level. In addition to online deployments, these
mass spectrometers are also used to analyze particles that were collected offline on filters (Siegel et al., 2021;Daellenbach et
al., 2016;Huang et al., 2019).
In this study, coupled offline filter collection done in Beijing in autumn 2018 and a FIGAERO high-resolution time-of-
flight chemical ionization mass spectrometer (FIGAERO-CIMS, Aerodyne Research Inc., US) to investigate (1) OA
composition at molecular level during different haze types and (2) its implications for aerosol light-absorptive properties.
**2. Method**
**2.1 Sampling information**
The sampling site (39° 56'31" N, 116°17'50" E) is located on the west campus of Beijing University of Chemical
Technology (BUCT), which is near the West Third Road in urban Beijing and surrounded by residential areas with local
pollution sources such as traffic, residential heating and cooking emissions. The site is located on the top floor of a five-floor
building, about 20 m above ground level. Detailed information on the sampling site and its characteristics are reported in
previous studies (Kontkanen et al., 2020;Liu et al., 2020b;Cai et al., 2020;Zhou et al., 2020;Kulmala et al., 2021;Yan et al.,
2021;Yao et al., 2020). During the sampling period (Nov 3 to Nov 16, 2018), particulate matter with a diameter of 2.5 µm or
less ($PM_{2.5}$) was collected on filters using a four-channel sampler (TH-16A, Tianhong Co., China) with a sampling flow rate
of 16.7 L min$^{-1}$. 12-h $PM_{2.5}$ nighttime (21:30-9:00, the next day) and daytime (9:30-21:00) samples were collected on 47 mm
quartz filters (7202, 47mm, Pall Corp., US), pre-baked for 4.5 hours at 550 °C before sampling. The pre-baking time was
selected following procedures in a previous study (Liu et al., 2016) to ensure the removal of potential organic contamination.
A total of 27 samples (the Nov 6[th] daytime filter was not analyzed due to a data acquisition error) and 3 blanks were collected
(sampling dates are shown in Figure 1 and Table S1). We also conducted detailed comparison between the quartz and Teflon
filter samples (presented in Cai et al. in preparation). After sampling, the filters were wrapped in aluminum foil, sealed in a
sealing bag and stored in a freezer at -20 °C until analysis.
**2.2 Offline FIGAERO-CIMS analysis**
The filters were analyzed using the FIGAERO-CIMS in offline mode (Cai et al., in preparation). In brief, we took punches
(2 mm in diameter) of the collected quartz filters and put them between two pre-baked originally sized (25 mm) Zefluor®
Teflon filters that fit the FIGAERO filter holder ("sandwich technique"). The particles collected on the filter punch were
thermally desorbed by high purity nitrogen gradually heated from room temperature to 200 °C. The desorbed molecules were
then charged by addition of iodide (I$^-$), which is formed via exposure of methyl iodide to a radioactive source, Po$^{210}$ in this
study (Lopez-Hilfiker et al., 2014). The total ion count (TIC) varied between ~600,000 and 1.2 million s$^{-1}$ during analysis. To
avoid depletion of the reagent ion by the large amount of gaseous HNO$_3$ evaporating even from the small pieces of filter
samples at heating temperatures between 80 and 100 °C, a non-uniform temperature ramping procedure was applied (Figure
S1): Samples were (1) heated from room temperature (~25 °C) to 60 °C in 8 min, (2) from 60 to 110 °C in 15 min, (3) from
110 °C to 200 °C in 12 min, and (4) held at 200 °C for an additional 20 min ("soak"). The analysis protocol, data analysis
flow and method characterization are detailed in Cai et al. (in preparation).
FIGAERO-CIMS data were analyzed with the Tofware package (v.3.1.0, Tofwerk, Switzerland and Aerodyne, US) within
the Igor Pro software (v.7.08, Wavemetrics, US). We identified the molecular composition of 946 ions in the $m/z$ range 46 to
500 Th. Most of them (939 ions) were clustered with I$^-$. The rest were 7 inorganic ions with low molecular weight (NO$_2^-$,
NO$_3^-$, HSO$_4^-$, HN$_2$O$_5^-$, NO$_6$S$^-$, H$_2$NO$_7$S$^-$, H$_2$N$_3$O$_9^-$) and not considered in the following discussions. Identified CHOX
compounds (compounds with molecular composition C$_{c\geqslant1}$, H$_{h\geqslant2}$, O$_{o\geqslant1}$, X$_{0-n}$, X can be N, S, or both) were grouped into (1)
compounds containing only carbon, hydrogen, and oxygen (CHO, 65±5% of total CHOX signal), (2) nitrogen-containing
compounds (CHON, 30±5%), (3) sulfur-containing compounds (CHOS, 5±1%), and (4) compounds containing both nitrogen
and sulfur (CHONS, 0.2±0.05%). The time series of the signal intensities of each compound during a heating cycle was
normalized to the signal of the reagent ion I$^-$. Backgrounds were determined using field blanks, which were scaled by the ratio


in signal during the last 1.5–3 min of the soak period of samples and field blanks to account for instrumental backgrounds. A
detailed discussion on background determination for offline FIGAERO data can be found in Cai et al. (in preparation). The
background-subtracted signal intensities over the entire heating cycle, which includes temperature ramp and soak, were
integrated, resulting in a single data point (in total ion counts) per compound and filter sample. Since in this study we focus
on the variability of the molecular composition of oxygenated OA and its relative changes, we did not attempt to convert total
ion counts into atmospheric concentrations. A discussion on the determination of sensitivity for the FIGAERO offline method
is presented in Cai et al. (in preparation).
In OA compound analysis, double bond equivalents (DBEs) provide information on the potential number of rings and double
bonds in a molecule. DBEs were calculated following the method proposed by Wang et al. (2017), shown as in Eq. (1):
$$DBEs = 1 + c - \frac{1}{2}h + \frac{1}{2}n \qquad (1)$$

where $c$, $h$ and $n$ are the number of C, H, and N atoms in the molecular formulae of the corresponding compounds.

**2.3 Collocated measurements and analyses**
An online Time-of-Flight-Aerosol Chemical Speciation Monitor (ACSM, Aerodyne Research Inc., US) equipped with a
$PM_{2.5}$ lens and standard vaporizer was operated at the same site. In this study, the ionization efficiency (IE, 230 ions $pg^{-1}$) and
relative ionization efficiencies (RIE) for $NH_4$ (4.0), $NO_3$ (1.05), $SO_4$ (0.86) and Cl (1.5) were determined by calibrations with
pure standards of ammonium nitrate, ammonium sulfate and ammonium chloride, while the RIE of OA (1.4) was taken from
the literature (Canagaratna et al., 2007). A composition-dependent collection efficiency (CE) for ACSM was applied
following the method proposed by Middlebrook et al. (2012). Organic carbon (OC) and elemental carbon (EC) of $PM_{2.5}$ were
measured by a semi-continuous OC/EC carbon aerosol analyzer (Model-4, Sunset Laboratory Inc. US) with a time resolution
of 1 hour. The instrument was routinely calibrated with a solution of sucrose.
Gaseous $NH_3$ was measured by a collocated nitrate Chemical Ionization–Atmospheric Pressure interface–Time Of Flight
mass spectrometer (nitrate CI-APi-TOF, Aerodyne Research Inc., US). Meteorological parameters, including temperature,
relative humidity (RH), wind direction and wind speed were measured at the same site. The boundary layer height was
calculated by the method proposed by Eresmaa et al. (2012) based on ceilometer (CL-51, Vaisala Inc.) measurements and
used to identify the stagnant conditions typical for haze episodes.
The aerosol water content (AWC) for the sampling period was calculated with ISORROPIA II (Fountoukis and Nenes, 2007)
based on the chemical composition of non-refractory $PM_{2.5}$ (NR-$PM_{2.5}$) measured by the ACSM, and gaseous $NH_3$.
ISORROPIA II was run in forward and metastable modes to achieve stable performance (Wang et al., 2020;Guo et al., 2017).
Although uncertainties can arise for aerosol pH calculations due to missing measurements (e.g. HCl and $HNO_3$ and water-
soluble metal cations), we consider our calculations to be robust as the AWC is dominated by RH, temperature and major
components of particles (Guo et al., 2017;Guo et al., 2015).
Aerosol light absorption measurements were conducted with a multi-wavelength aethalometer (Model AE-33, Magee
Scientific Co., US) equipped with a $PM_{2.5}$ cyclone. The aethalometer measures the optical attenuation (ATN) of light
transmitted through PM collected on filters at seven wavelengths (370, 470, 520, 590, 660, 880 and 950 nm) with a time
resolution of 5 min. To fill a data gap from Nov 3 to Nov 6 due to calibrations at the BUCT site, we also analyzed the data
from another AE-33 located at the Tower Branch of the Institute of Atmospheric Physics (IAP), Chinese Academy of Sciences.
The IAP site is located ~6 km northeast of the BUCT site. During the entire month of Nov, the BC analyses agreed well
between the two measurement locations ($r$ = 0.94–0.95 and intercept = 0.33–0.58 µg $m^{-3}$ for the 7 wavelengths, Figure S2).
**2.4 Aerosol optical properties calculations**
The light absorption coefficient ($b_{abs}$) is determined from the ATN measured by the aethalometer and corrected for the so-
called shadowing effect (Virkkula et al. (2015)), which represents attenuation variation due to high mass loadings on the filter.
BC mass concentrations are derived from the shadowing effect-corrected $b_{abs}$ (Hansen et al., 1983).
The variation of $b_{abs}$ as a function of wavelength ($\lambda$) is described by the Ångström exponent (AAE), which is typically
calculated using observations from a pair of wavelengths (Lack and Langridge, 2013) as in Eq. (2):






$$AAE = -\frac{\ln(b_{abs,\lambda_1}) - \ln(b_{abs,\lambda_2})}{\ln(\lambda_1) - \ln(\lambda_2)} \qquad (2)$$

In this study, we selected the two wavelengths of 370 nm ($\lambda_1$) and 880 nm ($\lambda_2$) from the aethalometer measurements to
calculate the AAE, following previous studies (Wang et al., 2018;Tao et al., 2020;Lim et al., 2014). It has been shown that in
contrast to BC, light absorption of BrC has a strong wavelength dependence, which results in high AAE values for BrC (4 to
7 (Cheng et al., 2016a)), and much lower AAE values for BC (0.8 to 1.1 (Teich et al., 2017)). An AAE value of 1.0 is generally
adopted for BC (AAE$_{BC}$, (Teich et al., 2017;Xie et al., 2019b;Cheng et al., 2016a) and also used in this study. Here we have
used these differences in AAE to separate $b_{abs}$ for BC and BrC following the method by Lack and Langridge (2013). Due to
the low absorption of BrC in the infrared and low concentrations of mineral dust in autumn Beijing (Zhang et al., 2013), it
can be assumed that $b_{abs}$ at 880 nm is only from BC particles. $b_{abs}$ at 370 nm for BC ($b_{abs,BC,370nm}$) and BrC ($b_{abs,BrC,370nm}$) can
then be calculated using Eqs. (3) and (4):
$$b_{abs,BC_{370\,nm}} = b_{abs,BC_{880nm}} \times \left(\frac{370}{880}\right)^{-AAE_{BC}} = b_{abs_{880nm}} \times \left(\frac{880}{370}\right) \qquad (3)$$

$$b_{abs,BrC_{370nm}} = b_{abs_{370nm}} - b_{abs,BC_{370nm}} \qquad (4)$$


We note that AAE$_{BC}$ can vary with many factors such as BC core size, coating thickness, morphology, etc.(Zhang et al.,
2018;Cheng et al., 2009); BC with a core-shell structure can have an AAE$_{BC}$ higher than 1.0 (Bond and Bergstrom, 2007).
We also calculated $b_{abs,\,BrC370nm}$ following the empirical equation method proposed by Wang et al. (2018) using Mie theory
calculation and observed a high correlation ($r = 0.98$ and intercept of 1.6 Mm$^{-1}$) of the time series between the two
aforementioned methods.
The contribution of BrC to total aerosol absorption at 370nm ($P_{BrC}$) is assessed by Eq. (5):
$$P_{BrC} = \frac{b_{abs,BrC_{370nm}}}{b_{abs_{370nm}}} \qquad (5)$$

Lack and Langridge (2013) postulated that using different values for AAE$_{BC}$ and AAE$_{BrC}$ to attribute aerosol light-absorption
to organic and black carbon, respectively, is only valid when there is substantial light absorption contribution ($P_{BC}>23\%$)
from BrC; the average $P_{BrC}$ in our study period is 34±9%.
The light absorption of BC can be enhanced due to the lensing effect (BC absorption enhancement $E_{abs}$); Jacobson et al.
(2001) reported factors of up to 2.9. $E_{abs}$ of BC was calculated here as the ratio of light absorption of BC particles measured
at 880 nm by the aethalometer to the theoretical absorption from uncoated pure BC at 880 nm (Eq. (6), (Zhang et al., 2018;Xie
et al., 2019a)). The latter is calculated by multiplying EC concentrations (measured by the OC/EC analyzer) by the pure BC
mass absorption coefficient (MAC, 7.5 m$^2$/g) taken from literature (Bond and Bergstrom, 2007;Wu et al., 2018).
$$E_{abs} = \frac{b_{abs,BC_{880nm}}}{b_{abs,pureBC_{880nm}}} = \frac{b_{abs_{880nm}}}{EC \times MAC_{pure,uncoated}} \qquad (6)$$


## 3. Results and discussion

### 3.1 Three haze episodes: Temporal variation of PM$_{2.5}$ components and meteorological conditions

During the period of sampling, we observed three particulate pollution or haze episodes (visibility <10 km and RH< 90%
(Cai et al., 2020)) with NR-PM$_{2.5}$+BC concentrations higher than 100 µg m$^{-3}$, Nov 3 to 4, Nov 7 to 9 and Nov 11 to 15 (Figure
1). Between these episodes, 12-h NR-PM$_{2.5}$+BC concentrations decreased to <15 µg m$^{-3}$. During the cleaner days (Nov 5 to
Nov 6 and Nov 9 to Nov 10), the OA mass spectra from FIGAERO-CIMS were generally similar (shown in Figure S3). We



selected the days of Nov 3 (Ep1), Nov 8 (Ep2), Nov 14 (Ep3) and Nov 10 (clean period) to compare the molecular composition
of OA and derive particle optical properties. Even though OA concentrations were similar (Ep1: 49 µg m$^{-3}$, Ep2: 30 µg m$^{-3}$,
Ep3: 40 µg m$^{-3}$), the AWC exhibited large differences (Ep1: 65 µg m$^{-3}$, Ep2: 12 µg m$^{-3}$, Ep3: 263 µg m$^{-3}$), indicative of
different haze formation mechanisms.
Figure 1 shows the time series of temperature, RH, simulated AWC, wind direction and wind speed, as well as the time
series of the chemical components during the sampling period. We observed strong diel patterns and a slightly decreasing
trend in temperature during the whole sampling period. The wind direction and wind speed did not strongly influence the
pollution levels, likely due to the on average relatively low wind speed (0.6 m/s). The ratio of $SO_4$ to $NO_3$ (Fig. 1d) was
0.47±0.45, much lower than in the year 2005 ($SO_4/NO_3$ = 1.6) in Beijing (Yang et al., 2011), illustrating that nitrate has
become a more important PM component due to $SO_2$ reductions in North China during the last decade. We multiplied the
CHOX signals from FIGAERO-CIMS with their corresponding molecular weight to present the total CHOX abundance.
Similar temporal variation was observed between CHOX abundance and the OA concentrations from ACSM ($r$=0.94, Figure
1(c)).
Ep1 and Ep3 were strong haze episodes, with hourly concentrations of PM$_{2.5}$ of over 200 µg m$^{-3}$ and high concentrations of
secondary inorganic aerosol (SIA) compounds such as nitrate, ammonium and sulfate. The amplitude of the diurnal cycles of
temperature and RH were reduced when NR-PM$_{2.5}$+BC concentrations were larger than 200 µg m$^{-3}$ in both episodes. The
highest hourly AWC was larger than 100 µg m$^{-3}$ and 400 µg m$^{-3}$ in Ep1 and Ep3, respectively. In addition to the similarly
high RH and AWC, Ep1 and Ep3 were both characterized by the strong influence of air masses arriving from the south of the
North China Plain (NCP) (Figure S4). Such conditions are typical for the most severe haze episodes observed in Beijing (Sun
et al., 2015;Sun et al., 2013), where high RH and AWC lead to heterogeneous processes and a strong increase of SIA. In Ep1
and Ep3, the increase of OA concentrations and $f_{44}$– the fraction of signal measured by ACSM at mass-to-charge ratio 44 and
an indicator of more oxygenated and thus secondary OA (Ng et al., 2011) - shows that not only secondary inorganic but also
secondary organic species contributed strongly to those two severe haze episodes (shown in Figure 1)). A complete buildup
process of haze was observed in the period of Nov 11 to 15 with Ep3, which seems to occur in two phases: Start of pollution
accumulation under relatively dry conditions (Nov 11 – Nov 13), and then the development of haze with high AWC (Nov 13
to Nov 14).
Ep2 (Nov 8) with the highest hourly PM$_{2.5}$ concentrations of 150 µg m$^{-3}$ was characterized by a prominent OA contribution
(43% of NR-PM$_{2.5}$+BC) as well as a higher OA to $NO_3$ ratio (1.5, Figure 1d) compared to Ep1 (24%, 0.50) and Ep3 (27%,
0.53), more similar to the cleaner periods during the whole sampling period with PM$_{2.5}$ <35 µg m$^{-3}$ (52%, 3.4). In addition,
AWC and RH were much lower during Ep2 than during Ep1 and Ep3. This indicates a different haze formation mechanism
governing Ep2 compared to Ep1 and Ep3.
The clean period (Nov 10) is characterized by low PM and AWC levels, with average PM$_{2.5}$ and OA concentrations of 14±7
µg m$^{-3}$ and 8.4 ±4 µg m$^{-3}$, respectively. These are much lower than the average values of the whole sampling period (76±79
µg m$^{-3}$ and 22 ±15 µg m$^{-3}$, respectively). During the clean period, the highest value of OA/$NO_3$ during the sampling period
was observed (>10), illustrating the rather small influence of SIA.

### 236  3.2 Molecular composition of OA

The three haze episodes varied in the relative contribution of OA to total NR-PM$_{2.5}$+BC, and in the ratio of OA to inorganic
species as exemplified by the OA/$NO_3$ ratio in Figure 1(d). In the following, we examine the molecular composition of OA
more closely for the three episodes and the clean period. Figure 2(a) shows the stacked time series of the organic compounds
identified by FIGAERO-CIMS and grouped according to their molecular composition into CHO, CHON, CHOS, and CHONS
compounds, with the sum of all compounds referred to as CHOX. The time series of the sum of the signal of the CHOX
compounds measured by the FIGAERO-CIMS correlates well with that of the OA mass concentrations measured by ACSM
($r$ = 0.95), which shows the robustness of our sampling and analysis method. CHO (65±5%) and CHON (30±5%) compounds
dominated the CHOX signal, even though the relative contributions of the different groups varied between the different
episodes. Ep1 and Ep3 showed a high relative contribution of CHO and CHOS compounds (68% and 6.8% for Ep1, and 72%
and 7.3% for Ep3, respectively), which can be associated with the rapid formation of oxygenated OA and organosulfates
during haze in Beijing (Wang et al., 2021a;Le Breton et al., 2018), and relatively low contribution of CHON compounds (28%
and 21% in Ep1 and Ep3, respectively). On the opposite, for the clean period, the relative contributions of CHO and CHOS





were lower (56 and 3.4%, respectively), and those of CHON compounds were increased by a factor of ~2 times (40%)
compared to Ep1 and Ep3. In Ep2, characterized by low AWC, the CHO compounds had strong signal contributions (73%),
similar to Ep1 and Ep3, but much lower contributions of CHOS (3.6%) and a similar contribution of CHON (23%) were
observed.
For a more detailed look at the molecular composition of compounds during the different episodes, we further subdivided
the compounds measured by FIGAERO-CIMS based on their number of carbon atoms per molecule (Figure 2b). In general,
during the period analyzed here, compounds with less than 10 carbons contributed most to the total CHOX signal (78%±7%).
Although $<C_{10}$ compounds were dominant, variation of different carbon number compounds was observed for the different
periods. In Ep1 and Ep3, the contribution of compounds with low carbon numbers ($C_{2-6}$) was 83% and 88%, respectively,
while in the clean period their fractions went down to 73%. The signal intensities of $C_{2-4}$ compounds were >20 times higher
in Ep1 and 3 than the clean period, which is likely related to aqueous phase formation of small molecules (e.g. dicarboxylic
acids), as indicated by their high correlation with AWC ($r = 0.86–0.91$). Those small compounds are typically assumed to be
formed in the aqueous phase since gas-particle partitioning theory would favor larger precursor ($>C_7$) SOA semi-volatile
products in the particle phase (Lim et al., 2010). Another indication of aqueous SOA formation in Ep1 and Ep3 are the $f_{44}$ and
$f_{43}$ ratios of ~0.14 and ~0.06, which are within the narrow range of aqueous OA ($f_{44}$: 0.09–0.16 and $f_{43}$: ~0.06) observed in a
previous study in Beijing (Zhao et al., 2019). In contrast, the relative contributions of $>C_{10}$ compounds were higher in the
clean period (36%, compared to Ep3 with 18%), likely attributable to the stronger relative contributions from combustion
emissions. In Ep2, $C_6$ compounds were strongly enhanced (30%) compared to the clean period (18%) and Ep3 (14%), which
we associate with organics emitted from biomass burning (discussed below).
In Figure 2b we also plot the O:C ratio of CHO group derived from FIGAERO-CIMS data. Similar to what was shown
previously for winter of Beijing (Hu et al., 2017;Sun et al., 2016), the bulk O:C generally followed the trend of total OA and
total CHOX signal, i.e. higher OA concentrations coincided with more oxygenated OA. The highest O:C values (0.6 to 0.7)
were observed during Ep1 and Ep3, while during the clean days, the O:C ratio went down to 0.4 to 0.5. The higher O:C ratios
during the haze periods were likely due to the enhanced contribution of SOA. An SOA component related to aqueous-phase
processes was found to be a dominant factor for the increase of the degree of oxygenation of OA during a humid pollution
period in Beijing (Sun et al., 2016;Zhao et al., 2019). In-cloud or droplet processes may be enhanced and form OA compounds
such as small acids (e.g.oxalate) (Guo et al., 2010) and humic-like substances (HULIS) (Laskin et al., 2015). We can therefore
expect that the compounds with small carbon numbers that show higher contributions during humid haze periods (e.g. Ep3)
may be carboxylic acids and therefore have a relatively high O:C ratio.
With secondary OA species being related to smaller carbon number, the temporal variation of the bulk average carbon
number was then similar to that of the BC fraction of total $PM_{2.5}$ ($f_{BC}$) (Figure 2c). BC is a typical indicator of primary
combustion emissions (residential heating, traffic exhaust) in Beijing (Cai et al., 2017;Cai et al., 2020;Sandradewi et al.,
2008;Zotter et al., 2017). Through secondary formation and oxidation reactions at a later stage of the haze between Nov 11
to Nov 15, the contribution of secondary components increased, resulting in a decrease of $f_{BC}$ and H:C ratios, while the O:C
ratio increased. In Ep2, BC and $f_{BC}$ increased to 10 µg m$^{-3}$ and 9.2% compared to <2 µg m$^{-3}$ and 3.1% on Nov 5 (clean day,
the end of Ep1), suggesting that this episode was more influenced by primary emissions rather than secondary formation.
Also, the signal of $C_6$ compounds was increased (shown in Figure 2(b)) due to the increase of $C_6H_{10}O_5I^-$, which corresponds
to anhydrous sugars such as levoglucosan, mannosan, galactosan, and 1,6-anhydro-β-D-glucofuranose from the breakdown
of cellulose during wood combustion (Simoneit et al., 1999)), tracers for biomass burning activities. Another indicator for
biomass burning, $f_{60}$, representing the fraction of $C_2H_4O^+$ to total OA (Cubison et al., 2011) measured by ACSM, was also
increased in Ep2. Ep2 was overall characterized by a larger influence of biomass burning emissions, which is not the case for
Ep1 and 3. In Figure S5, we further show the carbon number-segregated O:C ratios during the sampling period, which confirm
the different nature of haze episodes 1 and 3 compared to Ep2: the percentage contribution of $C_6$ compounds to CHO and
their O:C ratios were different during Ep2 (42% and 0.8, respectively), compared to 25% (19%) and 0.7 (0.7) for Ep1 (Ep3).
The respective roles of different processes such as gas-to-particle conversion and condensed-phase reactions in the increase
of OA mass and O:C ratio during the haze episodes can be investigated by looking at the mass increase of carbon, oxygen,
and hydrogen in the particle phase separately. As shown in Figure 2(d) and Figure S6, the signal-weighted mass (defined as
atom number multiplying their atomic mass) of elements C, H, O, N was generally increased during the three episodes, but
the increase in mass concentrations of OA was mainly driven by the addition of both carbon and oxygen, implying that
ageing/oxidation reactions (e.g. functionalization of particle-phase organics and aqueous-phase reactions), and gas-to-particle
conversion contributed to SOA formation in haze episodes.
We also calculated the relative atom fraction of the individual atoms of all CHOX compounds ($f_{atom signal}$) using Eq. (7):



$$f_{atom\ signal} = \frac{\sum Signal_i \times Atom_{i,j,num} \times AM_{i,j}}{\sum Signal_i \times MW_i} \qquad (7)$$

Where $Signal_i$ and $MW_i$, represent the signal intensity and molecular weight of compound $i$. respectively, and $Atom_{i,j,num}$ and
$AM_{i,j}$, the number and atomic mass of atom $j$ in compound $i$, respectively. The time series of $f_{atom\ signal}$ is shown in Figure 2
(e). Compared to the clean period, a higher $f_{atom\ signal}$ of O and slightly lower contributions of C and H were measured in Ep1
and Ep3. This indicates again that oxidation reactions play an important role in the increasing total OA mass in the humid
haze periods. We can, however, based on this analysis, not make any conclusions about the importance of aqueous-phase
reactions. The $f_{atom\ signal}$ of N decreased during the haze periods (Ep3: 21%, clean period: 29%) while S increased (Ep3: 1.4%,
clean period: 0.67%), consistent with the CHON and CHOS group fraction variations (shown in Figure 2(a)). Although the
mechanism of organosulfur and inorganic sulfate formation in heterogeneous reactions is not fully understood, it seems
probable that $SO_2$ is rapidly oxidized and sulfate/organosulfur is formed in aerosol water with different types of oxidants and
catalysts (Song et al., 2018;Cheng et al., 2016b;Liu et al., 2020a;Wang et al., 2021b).
In the following, we further characterize SOA in the different episodes with respect to: (1) compounds across different
carbon and oxygen numbers, (2) compounds with different DBEs and (3) homologous-like series and individual compounds
in typical episodes. The distribution of compounds with different carbon and oxygen numbers is shown in Figure 3. During
the clean period, the $C_6$ compounds, especially those with 5 oxygen atoms (and particularly $C_6H_{10}O_5I^-$), made up 5.9% of all
CHO compounds, 70% higher compared to Ep1 and Ep3. This indicates that biomass burning emissions played a relatively
more important role during clean periods. Another important characteristic of the clean period was the enhancement of the
relative contributions of nitrogen-containing organics, which were dominated by $C_6H_hO_3N_n$ compounds (possibly
nitrophenols, 5% to CHOX, shown in Figure S7).
During Ep2, the signals of compounds with 6 carbon and 5 oxygen atoms increased to up to 21% of total CHOX signals,
which is a much higher fraction compared to the clean period (5.9%) and haze days (3.5%). Also during Ep2, $C_6H_{10}O_5I^-$ was
the main contributor to this group (shown in Figure S3). The time series of $C_6H_{10}O_5I^-$ (as well as its fraction of CHOX,
$f_{C6H10O5I^-}$) follows the trend of $f_{60}$ measured by the ACSM, and both are strongly enhanced during Ep2 (shown in Figure 2 (b)
and Figure S8). The count median diameter (CMD) of the particles was around 60 nm (Figure S4) in Ep2, consistent with
fresh biomass burning emissions (50–70 nm during flaming (Vu et al., 2015)), in contrast to the CMD in the clean period (~20
nm) and haze episode Ep3 (~100 nm).
Ep1 and Ep3 were characterized by higher contributions of compounds with small carbon numbers (<6). Here we show that
$C_{2-6}$ compounds with 4 oxygen atoms made up 23% (Ep1) and 27% (Ep3) of the total CHOX, which is 2 and 3 times higher
than in Ep2 and the clean period, respectively. Since Ep1 and Ep3 were characterized by lower UVB radiation (shown in
Figure S4), high AWC and high $f_{44}$, this indicates further that these dominant OA compounds with 4 oxygen atoms are
dicarboxylic acids, likely formed in aqueous phase reactions. The absolute signal intensity of oxalic acid ($C_2H_2O_4I^-$) was 50
and 70 times higher in Ep1 and Ep3 than in the clean period, and a high correlation was observed between dicarboxylic acids
and AWC ($r$ = ~0.75 for different dicarboxylic acids). As $PM_{2.5}$ concentrations increased with RH and AWC in Ep3, the
concentrations of OA gradually increased from <10 µg m$^{-3}$ (daytime of Nov 11) to over 50 µg m$^{-3}$ (nighttime of Nov 13) and
the OA molecular composition changed as well. On Nov 11, the signals of $C_6$ and $O_5$ compounds were prominent (shown in
Figure S9), similar to Ep2 and clean periods. As pollution levels increased, on Nov 12, the contributions of $C_{2-6}$ and $O_4$
compounds strongly increased and the compound distribution became more similar to the haze period (Figure S9), indicative
of the important role of AWC in SOA and severe haze formation in Beijing.
In order to further characterize organic compounds detected by FIGAERO-CIMS, we plot the Van Krevelen (VK)
diagrams of CHO and CHON compounds in Figure 4 and Figure S10, respectively. Each dot in Figure 4 represents a measured
OA compound, which is color-coded by the calculated DBE and sized by the square root of its signal. During the clean period,
the OA components displayed a higher contribution of unsaturated species (DBEs≥6, 11% of total CHOX in the clean period,
compared to 7.4% in Ep3) with lower H:C and O:C ratios. Typically, compounds with a DBEs to carbon ratio higher than 0.7
are characterized as soot or oxidized polycyclic aromatic hydrocarbons (PAHs) (Cui et al., 2019). The relative contribution
of the compounds with carbon number ≥ 6 meeting that criterion was around 12% of the total CHOX signal in clean periods,
which was higher compared to Ep3 (7.0%). Figure 4 shows that the compounds with high DBEs generally have between 2
and 3 oxygen atoms, implying that they underwent some oxidation. Those compounds with characteristics representative of
oxidation products of aromatics (Molteni et al., 2018) exhibit stronger relative contributions during clean periods. For example,
the relative intensity of $C_6H_6O_2I^-$, a benzene ($C_6H_6$) oxidation product, was around 3 times higher during the clean period.
The relative signal of $C_7H_8O_2I^-$, formed from toluene ($C_7H_8$) oxidation, was 40 to 70% higher compared to the haze periods.





For the CHON compounds, both the relative contributions of $C_6H_5NO_3I^-$ (possibly nitrophenol) and $C_7H_7NO_3I^-$ (possibly
methyl nitrophenol) exhibited 2 to 3 times higher relative contributions during clean days than during the haze episodes. The
average UVB radiation intensity for the daytime of Nov10 was around 4 times higher than during the haze episodes (shown
in Figure S4), which might result in higher levels of OH radicals and a stronger photo-oxidative potential. In addition, the
ratio of the signal intensities of nitrophenol to nitrocatechol ($C_6H_5NO_4I^-$) in clean days was about 5 times higher than during
the polluted days of Ep1 and Ep3, also consistent with recent findings in Beijing that the elevated $NO_2$ during polluted days
promotes the formation of nitrocatechols over nitrophenols (Wang et al., 2019b).
Generally, for Ep2 we found a number of CHO and CHON compounds reported from laboratory wood-burning ageing
experiments and ambient environments strongly influenced by biomass burning emissions (Lin et al., 2012;Mohr et al.,
2013;Bertrand et al., 2018;Daellenbach et al., 2019) enhanced compared to the clean period, such as $C_6H_{10}O_5I^-$ (24 times),
$C_6H_5NO_4I^-$ (33 times) and $C_7H_7NO_4I^-$ (possibly methyl-nitrocatechol, 7.7 times) (shown in Figure S3). The 72-h back
trajectory (air mass retroplume) calculated for Ep2 shows an influence of southern areas at the receptor site, where residential
biomass burning emissions are abundant (Figure S4).
In Ep1 and 3, substantially higher absolute signals of inorganic ions were observed compared to Ep2 ($HNO_3I^-$: 4 times (Ep1)
and 3 times (Ep3), $SO_3I^-$: 39 times (Ep1) and >500 times (Ep3)) and the clean period ($HNO_3I^-$: 43 times (Ep1) and 27 times
(Ep3), $SO_3I^-$:>200 times (Ep1) and 700 times (Ep3)). As discussed previously, it is worth noting that during heavy haze (Ep1
and Ep3), the signals of $CH_4SO_3I^-$ and $C_2H_4SO_4I^-$ were much higher than during Ep2 ($CH_4SO_3I^-$: 2 times for Ep1 and Ep3,
$C_2H_4SO_4I^-$: 7 times (Ep1) and 8 times (Ep3)) and the clean period ($C_2H_4SO_4I^-$: 19 times for both Ep1 and Ep3, $C_2H_4SO_4I^-$: 46
times (Ep1) and 58 times (Ep3)). As shown in Figure 4, a homologous-like series of dicarboxylic acids ($C_nH_{2n-2}O_4$) and a
series of compounds with one more DBEs ($C_nH_{2n-4}O_4$) were enhanced in Ep1 and Ep3 compared to Ep2. Apart from oxalic
acid discussed previously, other dicarboxylic acids such as malonic acid ($C_3H_4O_4I^-$), succinic acid ($C_4H_6O_4I^-$), and glutaric
acid ($C_5H_8O_4I^-$) showed much higher (20–60 times) signals compared to the clean period. These findings show that during
humid haze in Beijing, a homologous series of dicarboxylic acids, likely formed in the aqueous-phase, may make up a
substantial fraction of the more oxygenated OOA (MO-OOA) found in previous studies (Sun et al., 2016). It is also interesting
to note that the OA components measured in Ep1 and Ep3 were very similar to those measured at Peking University (PKU),
Beijing in winter 2017 during a haze episode with similar $PM_{2.5}$ loadings (PKU: 188 μg m$^{-3}$) and RH levels (PKU: 74%)
(Figure S11, Zheng et al., (in preparation)).
In summary, the haze episodes during our sampling period can be classified by two different formation pathways: (1) mainly
influenced by relatively fresh biomass burning emissions under low RH with strong OA compound signals of levoglucosan,
aromatics and N-containing aromatics, (2) dominated by aqueous-phase reactions with high RH and air masses coming from
the south of the NCP with more oxygenated and low molecular weight OA such as dicarboxylic acids. In the next section, we
will investigate how the OA compounds formed in different haze types affect aerosol optical properties.

### 383   3.3 Influence of OA compounds on particle optical properties

#### 384   3.3.1 Temporal variation of $b_{abs}$ and $E_{abs}$

To investigate particle optical properties during the sampling period, we display the time series of AAE, the BrC absorption
coefficient, the ratio of BC to EC, and $E_{abs}$ calculated together with OA (Figure 5). The time series of BC and OA generally
follow each other, with a stronger diel variation of BC, especially during Ep1 and Ep3 (shown in Figure S4). AAE exhibited
an inverse correlation with OA during Ep1 and Ep3, but not Ep2 when biomass burning occurred. Although still higher than
during Ep1 and Ep3, the AAE decreased from the end of Ep2 (Nov 9) to the clean period of Nov 10. The average AAE during
our sampling period was 1.4, slightly lower than in winter in Beijing (1.6, (Xie et al., 2019b)), likely due to the lower
contribution of residential heating activates in autumn than in winter. The variation in AAE throughout the sampling period
reflects aerosol optical properties being influenced by the variation of sources, compounds, pollution levels and formation
pathways. The temporal evolution of the normalized (to OA) $b_{abs,BrC,370nm}$ is correlated with $f_{60}$ ($r$=0.65) and shows an
enhancement during Ep2 and decreases during Ep1 and Ep3 when aqueous-phase reactions may be important. It shows that
even though the total OA concentrations and $E_{abs}$ were strongly increased in Beijing during the humid haze, the light-
absorption ability of the OA compounds seemed to decrease.
During Ep1 and Ep3, $E_{abs}$ was higher than during Ep2, indicating that BC particles were more aged and more thickly coated
by organic and inorganic constituents (Figure S12). The lower $E_{abs}$ of the clean period, on the other hand, implies that BC
particles were more likely freshly emitted, and therefore less of a potential lensing effect could be observed. Ratios of $PM_{2.5}$
major components to EC were used in a previous study to investigate shell effects on BC particles and $E_{abs}$ (Zhang et al.,





2018). Here, we show $E_{abs}$ variation as a function of SIA, POA and SOA to EC ratios (Figure S12). POA and SOA were
estimated based on an empirical formula with $f_{44}$ and $f_{57}$ from ACSM measurements as input (Ng et al., 2011). Consistent
with earlier work conducted in Paris, France (Zhang et al., 2018), $E_{abs}$ was substantially enhanced with increasing SOA-to-
EC ratio (up to 16), while the increase as a function of POA-to-EC and SIA-to-EC ratios was less prominent (shown in Figure
S12). SOA thus has the potential to be a more effective shell for BC particles than SIA and POA. The similarity of the AAE,
$E_{abs}$ patterns and $b_{abs}$ of different wavelengths from 2 sites ~6 km apart (BUCT and IAP sites, Figure S2) implies that these
effects are likely to occur on a regional scale in Beijing. They suggest that light-absorption of BC and BrC particles can be
strongly affected by different OA components and that the OA compounds formed in the two haze types have different light-
absorption properties.

### 3.3.2 Correlations between optical parameters and OA compound signals

OA compounds and their potential optical effects are investigated with a correlation analysis in this study. In Figure 6, we
show the histograms of the correlation coefficients ($r$) between the OA compound signals (normalized by EC), $E_{abs}$ (Figure 6
(a)) and $b_{abs,BrC,370nm}$/$b_{abs,\ BC,370nm}$ (Figure 6 (c)). We normalized OA and $b_{abs,BrC,370nm}$ since BrC and BC could be co-varied due
to the same sources and the influence of meteorology. We selected the 20 OA compounds with the highest $r$ as "key
compounds" for $E_{abs}$ of BC and $b_{abs,BrC,370nm}$ for brown carbon light absorption, respectively. The key compounds for $E_{abs}$
generally exhibited relatively low DBEs ($2.3\pm1.3$ for the CHO group and $2.6\pm1.3$ for the CHON group) and high O:C ratios
($0.86\pm0.34$ for the CHO group). The much higher O:C ratio of those compounds compared to all CHO compounds ($0.48\pm0.31$)
indicates that highly oxygenated SOA plays an important role in BC lensing effects and $E_{abs}$ of BC. CHON with 2 to 4 DBEs
such as $C_nH_{2n-1}NO_3^-$ and $C_nH_{2n-3}NO_3^-$ (e.g. amine/amides, organonitrites as well as organonitrates) also exhibited a high
correlation with $E_{abs}$. Overall, low MW compounds, CHO with 4 or 5 oxygen atoms and CHON compounds with 3 to 5
oxygen atom, such as $C_3H_4O_4I^-$, $C_3H_6O_4I^-$, $C_5H_6O_4I^-$, $C_6H_{10}O_4I^-$, $C_3H_5NO_3I^-$, $C_2H_3NO_3I^-$, exhibited the highest correlation with
$E_{abs}$ at 880 nm, with $r$ of 0.66–0.76. Their time series were similar, with strong enhancement during Ep1 and Ep3 (Figure
S13). It has been suggested earlier that MO-OOA could be more important for the BC lensing effect than less oxygenated
OOA (LO-OOA) and POA (Zhang et al., 2018). Based on our results we conclude that those small compounds (e.g.
dicarboxylic acids) potentially act as important coating shells creating a strong light absorption enhancement for BC during
the humid haze events. It should be noted that OA compounds could be both internally or externally mixed with BC-containing
particles and thus, the identified OA compounds may not necessarily be coatings on BC particles. Yet, considering the large
proportion of BC-containing to total particles during the heating season (60–78%, (Chen et al., 2020)) as well as the large
proportion of organics in BC-containing particles in Beijing (60%, (Wang et al., 2019a)), these OA compounds are very likely
important components of the BC coating shells with a high potential to increase $E_{abs}$.
Compared to $E_{abs}$, the key compounds for $b_{abs,BrC}$ such as $C_8H_8O_2I^-$, $C_8H_8O_3I^-$, $C_5H_5NOI^-$ and $C_7H_7NO_4I^-$ in general exhibit
higher DBEs ($3.4\pm0.68$ for CHO group and $3.9\pm1.0$ for CHON group) and lower O:C ratios ($0.32\pm0.12$ for CHO group).
These compounds are likely oxidized aromatics and nitro-aromatics. Apart from the aromatic-like compounds, $C_6H_{10}O_5I^-$ (e.g.
levoglucosan) and $C_6H_{12}O_5I^-$ were also found to be moderately correlated with $b_{abs,BrC,370nm}$/$b_{abs,\ BC,370nm}$, likely due to their co-
variation with light-absorbing carbon from biomass burning emissions. The time series of the key compounds for $b_{abs}$ all
showed a large enrichment during Ep2 (shown in Figure S13), confirming that biomass burning-related organics (e.g.
aromatics) and N-containing organics (e.g. nitrophenol and nitrocatechol derivatives) were important contributors to the light
absorption by brown carbon. The correlation coefficient of the normalized OA compounds' signals and $b_{abs,BrC}$/$b_{abs,BC,370nm}$
was observed to be lower than the normalized signals with $E_{abs}$. The generally higher correlation for $E_{abs}$ is likely due to the
co-varied time series for OA components and $E_{abs}$ during the haze periods.
In summary, we presented a series of OA compounds that have the potential to influence OA light-absorption in two ways
in Beijing: (1) during humid haze, more oxygenated OA, with compounds such as dicarboxylic acids likely formed in aqueous
phase reactions, have the potential to strongly increase the absorption by BC due to the lensing effect, (2) during haze
dominated by fresh biomass burning emissions, compounds with a high number of DBEs and low O numbers, such as
aromatics and N-containing aromatics can act as brown carbon and potentially lead to more absorption at shorter wavelengths.

### 4. Conclusions

Although OA was found earlier to be one of the dominant factors for aerosol optical effects, the chemical composition of
OA may act in different roles in aerosol light absorption. To investigate the chemical composition of OA in a polluted





megacity and its effects on particle optical properties, in this study for the first time we relied on the molecular composition
of OA in autumn Beijing determined by FIGAERO-CIMS. We found that during severe humid haze periods, compounds with
a low number of DBEs and high O:C ratios (e.g. dicarboxylic acids) were strongly enhanced. In contrast, during a strong
biomass burning episode characterized by low AWC, compounds with a high number of DBEs and low O:C ratio were
observed. The comparison between low and high RH haze conditions indicates different mechanisms for haze formation in
Beijing, where the former was mainly influenced by local emissions while the latter was governed by secondary components
(potentially formed via aqueous-phase reactions) and more influenced by air masses from the southern NCP areas. This
implies that in order to reduce pollution in Beijing, the implementation of local direct particle emission control and gaseous
precursor emission control in the areas south of Beijing is necessary.
By combining the molecular composition of OA with aerosol light-absorption measurements, we found that the compounds
that are highly oxygenated, with a low number of carbon atoms and 4 oxygen atoms (e.g dicarboxylic acids) were strongly
increased during humid haze periods and highly correlated with $E_{abs}$. They are thus likely an important contributor to the
coating shells of BC particles and also a potentially important contributor of $E_{abs}$. Contrarily, the contribution of oxygenated
aromatics and nitro-aromatics were found to be closely linked to the light absorption of BrC.
In summary, we determined two kinds of haze episodes formed by different mechanisms in autumn Beijing: (1) driven by
high AWC and secondary formation, (2) driven by fresh emissions from biomass burning activities. We also determined the
OA molecular composition in those two types of episodes and in clean periods, which in turn influenced aerosol optical effects.
This is a step forward towards a better understanding of anthropogenic SOA formation in a highly-populated megacity, its
impacts on the local climate and its contribution to the air pollution cocktail.
*Author contributions*
MK, CM, KRD and JC designed the research. JC, CW, CM and KRD analyzed the FIGAERO-CIMS data. JC, JDW, XLF
and KRD analyzed the aethalometer data for the BUCT site. JDW and YLS provided aethalometer data for the IAP site. JC,
WD, FXZ, SH, XLF, BWC, LY, ZMF, TC, YCL, JTK, TP, JK, PC, DW, JZ, CY, FB, CM, MK and KRD performed the
online measurements and interpreted the results. JDW provided the emission inventory for North China and SH provided back
trajectory analysis. MK supported and supervised this research. JC, KRD, and CM wrote the manuscript with contributions
from all co-authors. All authors have given approval to the final version of this manuscript.
*Acknowledgements*
This work was supported by ACCC Flagship funded by the Academy of Finland (337549); "Quantifying carbon sink,
CarbonSink+ and their interaction with air quality" INAR project funded by Jane and Aatos Erkko Foundation; European
Research Council (ERC) with the projects ATM-GTP (nr. 742206) and CHAPAs (nr. 850614); Knut and Alice Wallenberg
Foundation (WAF project CLOUDFORM, grant no. 2017.0165). KRD acknowledges support by the SNF mobility grant
P2EZP2_181599





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



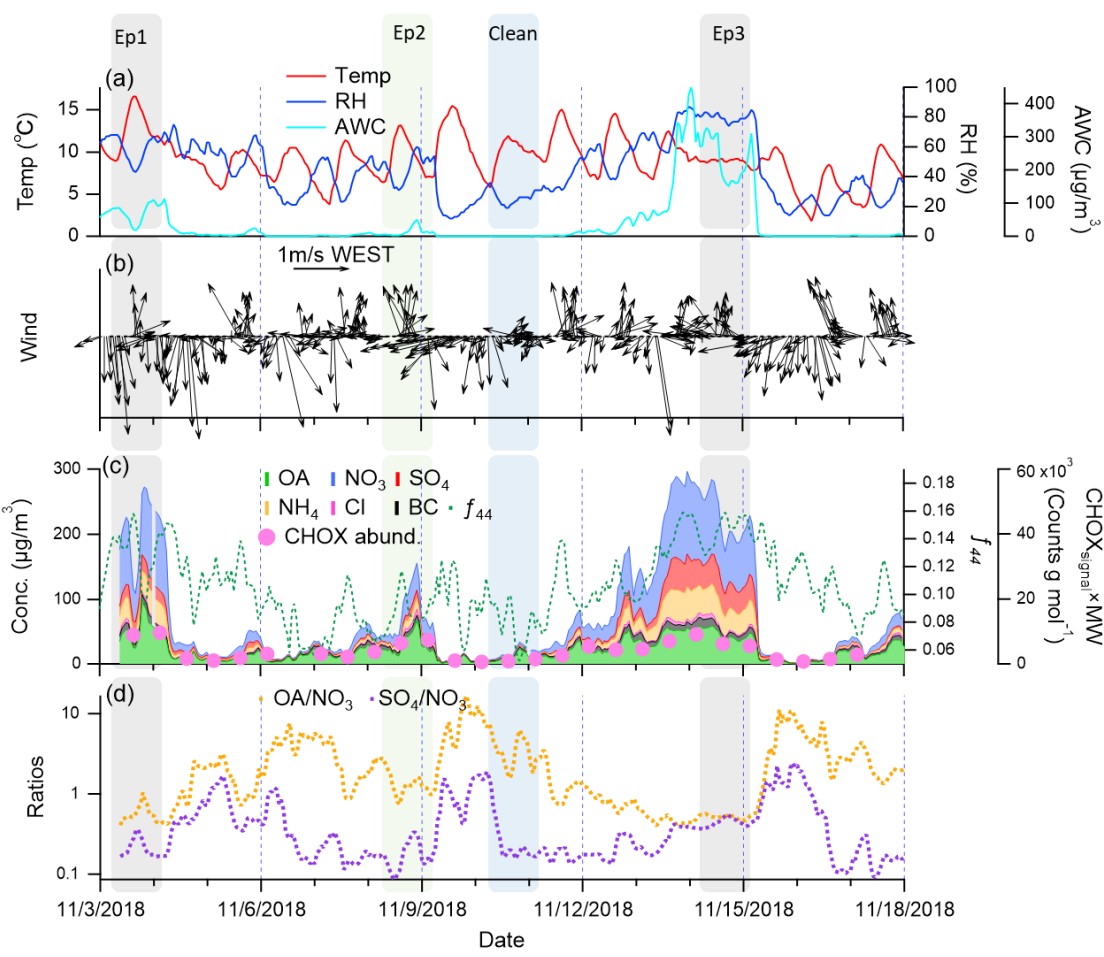


**Figure 1.** Time series of (a) temperature, relative humidity (RH), aerosol water content (AWC), (b) 1-hour averaged wind direction and wind speed, (c) chemical components of NR-PM$_{2.5}$, BC, $f_{44}$ from ACSM, CHOX abundance from FIGAERO-CIMS and their sampling dates are marked by pink dots, (d) OA/NO$_3$ and SO$_4$/NO$_3$.



**Figure 2**. Time series of (a) abundance of CHO, CHON, CHOS, CHONS compounds, and OA concentrations measured by ACSM (b) signals of compounds grouped according to carbon number, O:C ratio, (c) average carbon number, H:C ratio, the fraction of BC to NR-PM$_{2.5}$+BC, and $f_{60}$ from ACSM, (d) the signal-weighted total mass of elements C, O, H, N, S, and (e) the relative atom fraction of C, O, H, N and S.



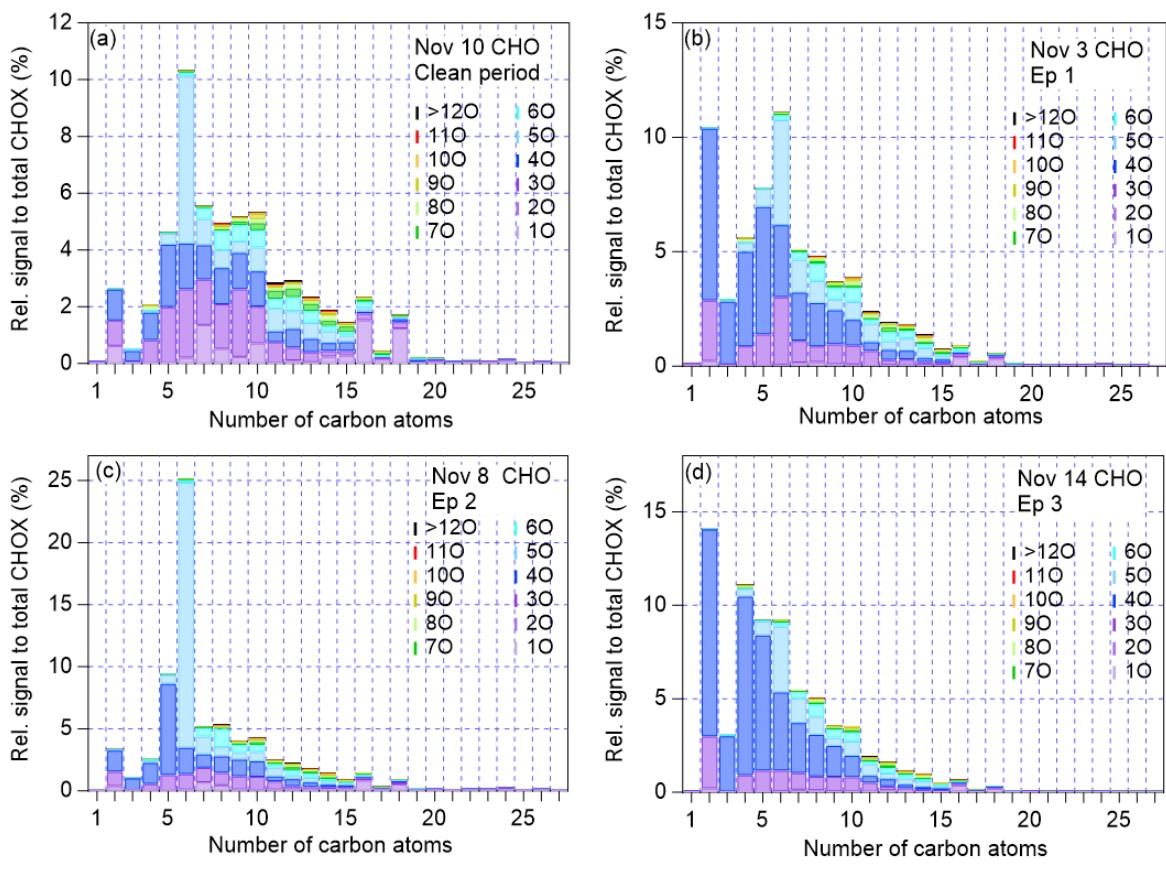

779

**Figure 3.** Signal fractions to total CHOX for CHO compounds with different numbers of oxygen and carbon atoms in (a) the clean period (Nov 10), (b) Ep1 (Nov 3), (c) Ep2 (Nov 8) and (d) Ep3 (Nov 14) periods. The same plots for CHON compounds are displayed in Figure S7.



783





784

**Figure 4.** (a) Van Krevelen (VK) diagram of CHO compounds in the clean period (Nov 10), (b) VK diagram of CHON compounds in the clean period (Nov 10), (c) VK diagram of CHO compounds Ep1 (Nov 3), (d) VK diagram of CHON compounds in Ep1 (Nov 3), (e) VK diagram of CHO compound in Ep2 (Nov 8), (f) VK diagram of CHON compound in Ep2 (Nov 8), (g) VK diagrams of CHO compound in Ep3 (Nov 14), (h) VK diagram of CHON compound in Ep3 (Nov 14). Each dot represents an identified compound with its H/C and O/C ratios and color-coded by its DBEs. O/C ratios in CHO and CHON groups calculated from the atom numbers in the formulae. The size of symbols is proportional to the square root of the relative signal intensity of each compound. The same plot color-coded carbon number is shown in Figure S10.

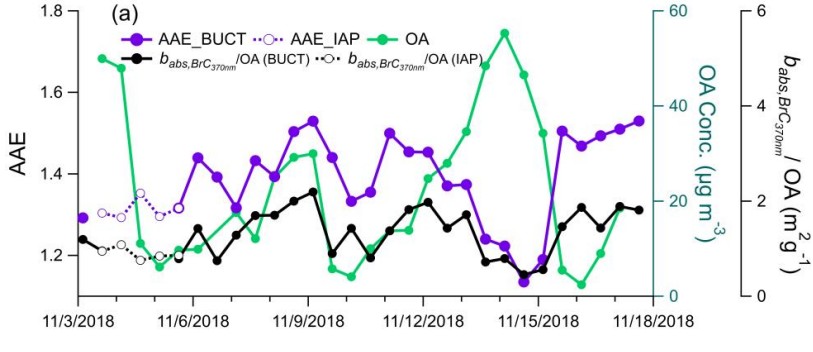

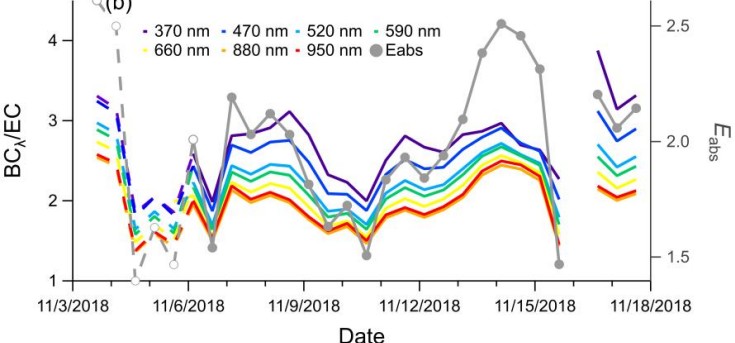

792

**Figure 5.** (a) Time series of AAE, normalized $b_{abs,BrC,370nm}$ (normalized to OA) and OA measured by ACSM during the sampling period, (b) ratio of BC to EC and $E_{abs}$ calculated with $BC_{880nm}$ and EC. The solid lines represent the parameters measured at the BUCT site and dashed lines represent the parameters measured at the IAP site.

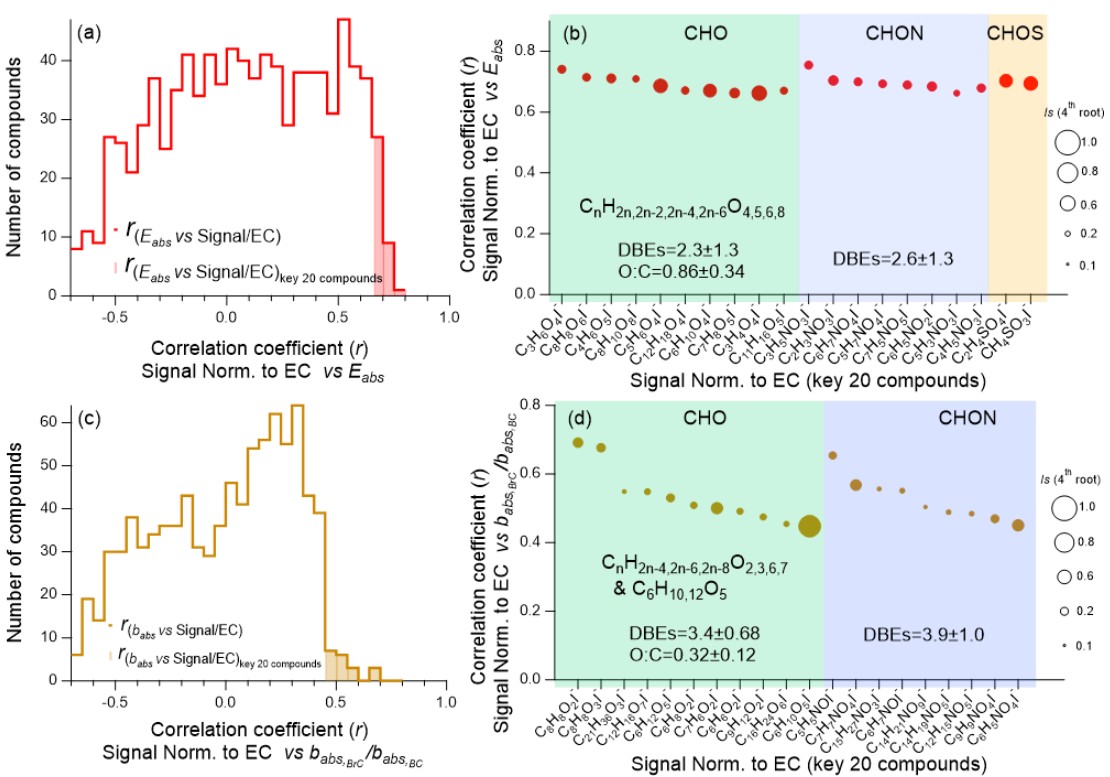

796

**Figure 6.** (a) Histogram of the correlation coefficients ($r$) between the normalized OA signals and $E_{abs}$ at 880 nm for all identified compounds (red line) and the key 20 compounds (red shaded area), (b) the correlation coefficients of key 20 compounds for $E_{abs}$ at 880 nm, (c) histogram of the correlation coefficients between the normalized OA compound signals and $b_{abs, BrC}/b_{abs, BC}$ at 370nm for all identified compounds (brown line) and the key 20 compounds (brown shaded area), and (d) the correlation coefficients of key 20 compounds for $b_{abs,BrC}/b_{abs,BC}$ at 370nm. The size of the symbols in (b) and (d) is proportional to the 4th root of the average signal intensities of the corresponding compound during the whole sampling period.

803

804