# Peer review of "Influence of organic aerosol composition determined by offline FIGAERO-CIMS on particle absorptive properties in autumn Beijing"

_Atmospheric Chemistry and Physics, 2021_

## Author Comment (AC1)

In this study, authors investigate chemical composition of particulate matter in Beijing, China and contrast these results with the measured aerosol optical parameters. The results of the manuscripts shed new light on how chemical composition influences the optical properties of the aerosol particles and introduces a new method for analyzing the collected particulate matter. The paper is overall well written and in scope of Atmospheric Chemistry and Physics journal but needs some corrections before publication.

**Reply: We are very grateful for the positive comments and helpful suggestions. We have carefully revised our manuscript accordingly.**

**Major comments:**

Section 2.2 (Offline FIGAERO-CIMS analysis): Major part of this section is left out to be detailed in future publication. Even though I understand the reasoning behind the decision, I feel that too many details are missing. In this state I feel that I cannot assess the credibility of the method and hence the results shown in the paper. The authors should present the draft of this future publication showing the main details of the methodology, or the used methodology should be presented in more detail in the current manuscript or in its supplement.

Reply: We realize that how we formulated this part of the method descriptions in the submitted manuscript may have been slightly misleading. The FIGAERO-CIMS has been used in offline mode previously (Huang et al., 2019a; Siegel et al., 2021), and the future publication mentioned merely provides an in-depth characterization and recommendations for best practices of this method. Here we largely followed the methodology used in those previous studies but needed to make adjustments to reduce reagent ion depletion due to high filter mass loadings. This is now explained in the SI as following, together with more information on how we determined backgrounds:

"Information on the offline FIGAERO-CIMS method can be found in previous studies (Siegel et al., 2021; Huang et al., 2019b). However, due to high mass loadings on our filters, we had to adjust the analytical protocol as follows:

 "sandwich technique" to be able to only use a small punch of the filter. We took 2mm punches of our filter samples and put them between two clean pre-baked (at 200 °C for 1 hour prior to usage) originally sized (25 mm) Zefluor® Teflon filters that fit the FIGAERO filter holder. This allows to reduce the amount of desorbed PM and thus to control reagent ion depletion (shown in Figure S1).

Figure R1 (S1). Schematic of the "sandwich technique" sample preparation

2) "non-uniform temperature ramping" protocol during the FIGAERO-CIMS desorption to reduce the rate of HNO3 vaporization and thus HNO3 signal and reagent ion depletion at temperatures between 80 and 100 °C, as following: (1) heating from room temperature (~25 °C) to 60 °C in 8 min, (2) from 60 to 110 °C in 15 min, (3) from 110 °C to 200 °C in 12 min, and (4) held at 200 °C for an additional 20 min ("soak") (shown in Figure S2).

Figure R2 (S2). The FIGAERO-CIMS temperature ramping protocol applied in this study

Background subtraction method to estimate instrumental and field blanks: The variation in instrument background due to the variation in mass loading is taken into account using the signal at maximum heating temperature (200 °C) and thus elevated temperature of surfaces downstream the filter. Thus, the total background signals are the field blanks (average of the 3 blanks) scaled by the signal ratios of ambient sample to blanks of the last 1.5–3 min of the soaking period."

We assessed the performance of the analytical protocol used by comparing the identified inorganic and total identified organic compounds (CHOX) by offline FIGAERO-CIMS (12-h samples) to the mass concentrations of  $PM_{2.5}$  chemical components measured by the co-located ACSM (Figure R3 (new Figure S3)). It shows the time series of the integrated signals of CHOX and inorganic component (HNO3I- and SO3I-) highly correlated with the OA, NO3 and SO4 concentrations from ACSM (Fig. R3, *r*=0.94–0.95).

The main text was also modified and expanded as follows:

"The filters were analyzed using the FIGAERO-CIMS in offline mode, largely following the approach proposed in previous offline FIGAERO-CIMS analyses (Siegel et al., 2021; Huang et al., 2019b). The particles collected on the filter were thermally desorbed by high purity nitrogen gradually heated from room temperature to 200 °C. The desorbed molecules were then charged by the addition of iodide (I-), which is formed via exposure of methyl iodide to a radioactive source,  $Po^{210}$  in this study (Lopez-Hilfiker et al., 2014). The IMR pressure was ~100 mbar and the total ion count (TIC) varied between ~600,000 and 1.2 million counts per second (cps) during analysis. Mass accuracy is within 10 ppm and the mass resolution is between 5000 to 6000 for ions>200 Th. In order to reduce reagent ion depletion, we adapted the analytical protocol as following: 1) we used a "sandwich technique" to hold small punches (2 mm in diameter) of the collected quartz filters (shown in Figure S1), which allowed reduction of the amount of measured PM2.5, 2) we used a non-uniform heating protocol for the FIGAERO-CIMS desorption: a slower temperature ramping rate was applied at heating temperatures between 80 and 100 °C to avoid depletion of the reagent ion by the large amount of gaseous HNO3 evaporating (shown in Figure S2 and S3). More information on the offline method including background determination can be found in the SI." (Line 102–113)

"The good correlation between FIGAERO-CIMS and ToF-ACSM (CHOX from FIGAERO-CIMS vs OA from ToF-ACSM, HNO3I- vs NO3 from ToF-ACSM, SO3I- vs SO4 from ToF-ACSM, see Figure S3) validates the offline FIGAERO-CIMS analyses – at least in terms of bulk PM constituents – and suggests that artefacts related to the method only play a minor role." (Line 126–129)

**Figure R3 (S3).** The comparison between signals from FIGAERO-CIMS and the concentrations of major components of  $PM_{2.5}$ : (a) CHOX versus OA from ACSM (b) HNO3I- versus NO3 from ACSM, and (c) SO3I- versus SO4 from ACSM

Line 346: Recent paper (Yang et al., 2021) showed that many compounds with high DBE (>2) and oxygen number (>4) are prone to thermal decomposition during FIGAERO heating. Lot of compounds shown in Figure 4 fall into this category, and I would not be surprised that some of the observed compounds could be thermal decomposition products. Authors should consider this possibility and discuss its implications to their results and conclusions.

Reply: We thank the reviewer for pointing out this new publication. We had noticed the recent results on FIGAERO thermal decomposition and double-checked our results accordingly. We would like to point out that the vast majority of signal in our study is from organic compounds with Onum<5 in both haze and clean periods (Figure R4). A certain degree of thermal decomposition is inevitable for some compounds during the FIGAERO thermal desorption process, mostly for highly functionalized and multifunctional OA compounds (Yang et al., 2021; Stark et al., 2017; Lopez-Hilfiker et al., 2016). However, the high abundance of compounds with DBEs <=2, and Onum

Figure R4 (revised from Figure 3). Signal fractions to total CHOX for CHO compounds with different numbers of oxygen and carbon atoms in (a) Ep3 (Nov 14) periods, (b) the clean period (Nov 10).

We further investigated the possibility of thermal decomposition through the shape of thermograms of these compounds. As thermal fragmentation commonly occurs at temperatures higher than thermal

desorption (Buchholz et al., 2020), we expect to see  $\geq$  2 modes in the thermogram for an ion that is also produced by the thermal decomposition of larger compounds. In Figure R5 (added to the SI as new Figure S18), we present the thermograms of the 10 most abundant OA compounds during the whole sampling period. Note that for thermograms resulting from the non-uniform heating protocol the signal was re-girdded to the temperatures of the uniform temperature ramping protocol. Most of these compounds (7 of 10) show a dominant single-mode in the thermogram and therefore are assumed not to be strongly affected by the thermal decomposition of larger compounds. Two compounds (C5H6O4I- and C6H8O4I-) showed two modes in their thermograms with higher signals for the first mode. Only one compound (C2H4O3I) seemed to be mostly a thermal fragment, as its signal is strongly dominated by the second mode at the higher temperature. Based on similar checks (Figure R7), we estimated that for the key compounds for light absorption (Figure 6), most of the small compounds (Cnum<6) in the CHO (72%, 5/7), and CHON or CHOS (73%, 8/11) groups are not significantly influenced by thermal desorption. For five compounds (C4H6O5I-, C5H12O5I-, C5H3NO3I-, C4H5NO3I-, C6H5NO4I-), however, the signal can be influenced by thermal desorption due to their relatively strong second thermogram peaks. We have also slightly revised Figure 6 (Figure R6) in the manuscript and marked those species by \* in the axis labels.

---

## Author Comment (AC2)

In this manuscript, Cai and colleagues discuss their findings on the correlation between the chemical composition and optical properties of organic aerosols based on online Aethalometer and offline FIGAERO CIMS measurements. A discussion on different haze formation mechanisms observed in Beijing is also presented. In general, the manuscript is well written and the discussion is easy to follow. With some necessary clarifications and discussions as outlined below, the manuscript would be suitable for publication in ACP.

Reply: We are very grateful for the positive comments and helpful suggestions. We have carefully revised our manuscript accordingly.

Offline FIGAERO-CIMS analysis: I agree with the other referee that more information is needed to validate the method. In addition to thermal decomposition, offline FIGAERO analysis would be susceptible towards sample handling and storage artefacts. For instance, acid-catalyzed reactions such as organonitrate hydrolysis (leaving CHO and HNO3) may occur on the filter. Would the small carboxylic acids desorb during storage? The "sandwich" technique for filter analysis may complicate quantification and/or volatility analysis due to increased vapor-filter interactions (there are now three filters as opposed to the single filter used for online FIGAERO).

Reply: We have added more information regarding the technique, thermal decomposition, and artefacts to the revised manuscript and SI. We refer the reader to the responses to reviewer 1´s comments on these topics. Whereas we believe that storing the filters frozen will minimize the reactions mentioned by the reviewer, we can only use the ACSM as a reference method, which quantifies OA composition at albeit lower chemical resolution. The good relative agreement of ACSM and FIGAERO-CIMS at bulk level (Figure S3), and the fact that we do observe time-dependent differences in the contribution of small carboxylic acids (Figure 4 and Table S2) in accordance with different haze conditions, gives us the confidence to declare our results as sound. Whereas we agree that the sandwich technique may complicate volatility analysis, which is not part of the present analysis, the integrated signal should not be affected as with the soak period compounds have enough time (20 min) to evaporate even from the more complex matrix of the sandwiched filters.

Here we present the additional information of the method that has been added in the revised main text and SI

Main text:

"The filters were analyzed using the FIGAERO-CIMS in offline mode, largely following the approach proposed in previous offline FIGAERO-CIMS analyses (Siegel et al., 2021; Huang et al., 2019). The particles collected on the filter were thermally desorbed by high purity nitrogen gradually heated from room temperature to 200 °C. The desorbed molecules were then charged by the addition of iodide ($I^-$), which is formed via exposure of methyl iodide to a radioactive source, $Po^{210}$ in this study (Lopez-Hilfiker et al., 2014). The IMR pressure was ~100 mbar and the total ion count (TIC) varied between ~600,000 and 1.2 million counts per second (cps) during analysis. Mass accuracy is within 10 ppm and the mass resolution is between 5000 to 6000 for ions>200 Th. In order to reduce reagent ion depletion, we adapted the analytical protocol as following: 1) we used a "sandwich technique" to hold small punches (2 mm in diameter) of the collected quartz filters (shown in Figure S1), which allowed reduction of the amount of measured $PM_{2.5}$, 2) we used a non-uniform heating protocol for the FIGAERO-CIMS desorption: a slower temperature ramping rate was applied at heating temperatures between 80 and 100 °C to avoid depletion of the reagent ion by the large amount of gaseous $HNO_3$ evaporating (shown in Figure S2 and S3). More information on the offline method including background determination can be found in the SI." (Line 102–113)

"The good correlation between FIGAERO-CIMS and ToF-ACSM (CHOX vs OA from ToF-ACSM, $HNO_3I^-$ vs $NO_3$ from ToF-ACSM, $SO_3I^-$ vs $SO_4$ from ToF-ACSM, see Figure S3) validates the offline FIGAERO-CIMS analyses – at least in terms of bulk PM constituents – and suggests that artefacts related to the method only play a minor role." (Line 126–129)

SI:

"Information on the offline FIGAERO-CIMS method can be found in previous studies (Siegel et al., 2021; Huang et al., 2019). However, due to high mass loadings on our filters, we had to adjust the analytical protocol as follows:

1) "sandwich technique" to be able to only use a small punch of the filter. We took 2mm punches of our filter samples and put them between two clean pre-baked (at 200 °C for 1 hour prior to usage) originally sized (25 mm) Zefluor® Teflon filters that fit the FIGAERO filter holder. This allows to reduce the amount of desorbed PM and thus to control reagent ion depletion (shown in Figure S1).

[Figure]

**Figure R1 (S1).** Schematic of the "sandwich technique" sample preparation

2) "non-uniform temperature ramping" protocol during the FIGAERO-CIMS desorption to reduce the rate of $HNO_3$ vaporization and thus $HNO_3$ signal and reagent ion depletion at temperatures between 80 and 100 °C, as following: (1) heating from room temperature (~25 °C) to 60 °C in 8 min, (2) from 60 °C to 110 °C in 15 min, (3) from 110 °C to 200 °C in 12 min, and (4) held at 200 °C for an additional 20 min ("soak") (shown in Figure S2).

[Figure]

**Figure R2 (S2).** The FIGAERO-CIMS temperature ramping protocol applied in this study

Background subtraction method to estimate instrumental and field blanks: The variation in instrument background is taken into account using the signal at maximum heating temperature (200 °C) and thus elevated temperature of surfaces downstream the filter. Thus, the total background signals are the field blanks (average of the 3 blanks) scaled by the signal ratios of ambient sample to blanks of the last 1.5–3 min of the soaking period."

[Figure]

**Figure R3 (S3).** The comparison between signals from FIGAERO-CIMS and the concentrations of major components of PM$_{2.5}$: (a) CHOX versus OA from ACSM (b) HNO$_3$I$^-$ versus NO$_3$ from ACSM, and (c) SO$_3$I$^-$ versus SO$_4$ from ACSM

Line 257-262: Do low-carbon compounds (e.g. C2-6 or C2-4) compounds have thermograms commensurate with their expected volatility? In other words, do they behave like "real" compounds, decomposition products, or a mixture of both during FIGAERO desorption? Some examples should be provided.

Reply: Thanks for the suggestion. In the revised supplementary information, we have added the thermograms of the most abundant compounds as suggested and also marked the compounds that may be affected by thermal decompositions in Figure 6.

And we have answered Reviewer #1's comment on the thermal decomposition. We also present it here.

We had noticed the recent results on FIGAERO thermal decomposition and double-checked our results accordingly. We would like to point out that the vast majority of signal in our study is from organic compounds with O$_{num}$<5 in both haze and clean periods (Figure R4). A certain degree of thermal decomposition is inevitable for some compounds during the FIGAERO thermal desorption process, mostly for highly functionalized and multifunctional OA compounds (Yang et al., 2021; Stark et al., 2017; Lopez-Hilfiker et al., 2016b). However, the high abundance of compounds with DBEs <=2, and Onum < 4 such as aldehydes, acids, especially monoacids, and diacids, has been found earlier for Beijing autumn- and wintertime with methods that do not include thermal desorption such as water-extraction with gas chromatography-FID (GC-FID)/ion chromatography (IC) or organic solvent extraction with gas chromatography chromatography-triple quadrupole mass spectrometry (GC/MS/MS) in Beijing (Guo et al., 2010; Zhao et al., 2018; Yu et al., 2021). It is therefore highly likely that those small compounds are formed by aqueous-phase reactions in winter Beijing as stated in this manuscripts and earlier publications (Lim et al., 2010; Zhao et al., 2018).

[Figure]

**Figure R4 (revised from Figure 3).** Signal fractions to total CHOX for CHO compounds with different numbers of oxygen and carbon atoms in (a) Ep3 (Nov 14) periods, (b) the clean period (Nov 10).

We further investigated the possibility of thermal decomposition through the shape of thermograms of these compounds. As thermal fragmentation commonly occurs at temperatures higher than thermal desorption (Buchholz et al., 2020), we expect to see $\geq 2$ modes in the thermogram for an ion that is also produced by the thermal decomposition of larger compounds. In Figure R5 (added to the SI as new Figure S18), we present the thermograms of the 10 most abundant OA compounds during the whole sampling period. Note that for thermograms resulting from the non-uniform heating protocol the signal was re-girdded to the temperatures of the uniform temperature ramping protocol. Most of these compounds (7 of 10) show a dominant single-mode in the thermogram and therefore are assumed not to be strongly affected by the thermal decomposition of larger compounds. Two compounds ($C_5H_6O_4I^-$ and $C_6H_8O_4I^-$) showed two modes in their thermograms with higher signals for the first mode. Only one compound ($C_2H_4O_3I^-$) seemed to be mostly a thermal fragment, as its signal is strongly dominated by the second mode at the higher temperature. Based on similar checks (Figure R7), we estimated that for the key compounds for light absorption (Figure 6), most of the small compounds ($C_{num} \leq 6$) in the CHO (72%, 5/7), and CHON or CHOS (73%, 8/11) groups are not substantially influenced by thermal desorption. For five compounds ($C_4H_6O_5I^-$, $C_5H_{12}O_5I^-$, $C_5H_3NO_3I^-$, $C_4H_5NO_3I^-$, $C_6H_5NO_4I^-$), however, the signal can be influenced by thermal desorption due to their relatively strong second thermogram peaks. We have also slightly revised Figure 6 (Figure R6) in the manuscript and marked those species by * in the axis labels.

[Figure]

**Figure R5 (Figure S18).** Normalized thermograms of the ions of (a) $C_2H_2O_4I^-$, $C_3H_4O_4I^-$, $C_4H_6O_4I^-$, $C_5H_8O_4I^-$, $CH_4SO_3I^-$, $C_6H_5NO_3I^-$, $C_7H_7NO_3I^-$ and (b) $C_5H_6O_4I^-$, $C_2H_4O_3I^-$, $C_6H_8O_4I^-$, on Nov 14. The thermograms were normalized by the maximum signals during the desorption.

[Figure]

**Figure R6 (Figure 6).** (a) Histogram of the correlation coefficients ($r$) between the normalized OA signals and $E_{abs}$ at 880 nm for all identified compounds (red line) and the key 20 compounds (red shaded area), (b) the correlation coefficients of key 20 compounds for $E_{abs}$ at 880 nm, (c) histogram of the correlation coefficients between the normalized OA compound signals and $b_{abs, BrC}/b_{abs, BC}$ at 370nm for all identified compounds (brown line) and the key 20 compounds (brown shaded area), and (d) the correlation coefficients of key 20 compounds for $b_{abs,BrC}/b_{abs,BC}$ at 370nm. The size of the symbols in (b) and (d) is proportional to the 4th root of the average signal intensities of the corresponding compound during the whole sampling period. Compounds that possibly have a substantial contribution of larger thermally fragmented parent compounds are marked with * in the axis labels.

[Figure]

**Figure R7 (Figure S19).** Normalized thermograms of the ions of the 18 key compounds on Nov 14 with $C_{num} \leq 6$ in Figure 6 (a) CHO group compounds without strong influence by thermal decompositions, and (b) CHO group compounds with a potentially strong influence by thermal decompositions, (c) CHON group compounds without strong influence by thermal decompositions and (d) CHON group compounds a potentially strong influence by thermal decompositions. The thermograms were normalized by the maximum signals during the desorption.

The main text in the revised manuscript was modified as follows:

Main text

"Chemical characterization by FIGAERO-CIMS, essentially a thermodesorption technique, is prone to thermal decomposition. For example, more oxygenated multi-functional organic compounds such as citric acid ($C_6H_8O_7$) and sucrose ($C_{12}H_{22}O_{11}$) were found to be affected by thermal decomposition in the FIGAERO-CIMS (Yang et al., 2021; Stark et al., 2017). Since thermal decomposition generally occurs at temperatures higher than the desorption temperature of most compounds (Buchholz et al., 2020), multi-modal thermogram shapes can be used as an indicator for signal contributions from thermally fragmented compounds. Based on such analysis of the filter collected on Nov 14, among the 10 ions with the highest intensity, only one ($C_2H_4O_3I^-$) appeared to be affected strongly by thermal decomposition." (Line 136–142)

"Among those key species, in Figure 6 we marked the compounds that, according to their thermograms, likely are influenced by thermal decomposition ($C_4H_6O_5I^-$, $C_5H_{12}O_5I^-$, $C_5H_3NO_3I^-$, $C_4H_5NO_3I^-$, $C_6H_5NO_4I^-$). Most of the other small compounds ($C_{num}<6$) in the CHO (72%, 5/7), and CHON or CHOS (73%, 8/11) groups are not significantly influenced by thermal desorption." (Line 450–453)

Aerosol water content, Line 146 -151: How sensitive are the ISRROPIA results to ACSM measurements, e.g. contribution to NO3 by orgnaonitrates? Furthermore, detection of HCl and to some extent HNO3 should be possible with the NO3- CI-APi-TOF, and can further constrain the AWC estimations.

Reply: Thanks for the suggestion. We added the following discussion to the supplementary information.

"We assessed the uncertainty of AWC estimated by ISORROPIA II through a series of three sensitivity tests:

1) HCl: This sensitivity test uses measured gaseous HCl concentrations for estimating AWC. During the sampling period, gaseous HCl concentrations were measured by $NO_3$-CI-APi-TOF. The normalized HCl signals were calibrated by a comparison between $NO_3$-CI-APi-TOF and a co-located Monitor for AeRosols and Gases in Ambient air (MARGA; Metrohm Inc., Switzerland) after our sampling period (Sep 2 to Sep 6, 2019). The details for HCl measurements and calibration are presented in Fan et al. (2021).

2) HCl+Organonitrates: This sensitivity test is based on the HCl sensitivity test but also accounts for the contribution of particulate organic nitrates (PON) when calculating AWC. The fraction of PON to organic aerosol (OA) is estimated at 14.8% in Beijing during wintertime by a thermodenuder–aerosol mass spectrometer method (Xu et al., 2021). This fraction is also consistent with previous studies using the $NO^+/NO_2^+$ ratio from the AMS (Farmer et al., 2010). Thus, we used a PON/OA fraction of 14.8% and measured OA concentrations by ToF-ACSM to estimate the concentration of PON during our sampling period. Inorganic nitrate is calculated by subtracting PON from the total $NO_3$ measured by ToF-ACSM. Such an assumption would provide us an upper limit contribution of nitrate in PON since the mass contribution from other elements (e.g. C and H) in PON was also subtracted and further upper estimation of the AWC bias due to the nitrate in PON.

3) HCl+Organonitrates+Gaseous HNO$_3$: This sensitivity test is based on the HCl+Organonitrates sensitivity test but also accounts for gaseous $HNO_3$ when estimating AWC. Under the meteorological conditions during our sampling period ($RH_{avg}$=45%, $Temp_{avg}$= 9.2 °C), the particle-phase fractions ($\varepsilon$) of $NO_3$ ($\varepsilon(NO_3^-) = (NO_3^-/(HNO_3+NO_3^-))$, mol/mol)) is generally over 82% in autumn Beijing (Ding et al., 2019). The effects of gaseous $HNO_3$ on AWC are estimated with the assumption of $\varepsilon(NO_3^-)$ = 82% and PON/OA = 14.8%.

The AWC estimated in these three sensitivity tests agreed well with each other as well as with the base case used in the manuscript ($r^2$=1.0 and slope=1.0, shown in Figure R8). Therefore, we can safely draw the conclusion that gaseous HCl, HNO$_3$, and organonitrates do not significantly affect the AWC estimations in this study.

[Figure]

**Figure R8 (Figure S5)**. (a) Time series of AWC concentrations in the base case (AWC$_{base}$) and the three sensitivity tests: 1) including gaseous HCl (AWC$_{HCl(g)}$), 2) including gaseous HCl and organonitrate effects (AWC$_{HCl(g)+PON}$), 3) and including gaseous HCl, HNO$_3$, and organonitrate effects (AWC$_{HCl(g)+ HNO3(g)+PON}$), and (b) the correlation between the AWC concentrations in different cases

We refer to this discussion in the main text:

"The aerosol water content (AWC) for the sampling period was calculated with ISORROPIA II (Fountoukis and Nenes, 2007) based on the chemical composition of non-refractory PM$_{2.5}$ (NR-PM$_{2.5}$) measured by the ToF-ACSM, and gaseous NH$_3$. ISORROPIA II was run in forward and metastable modes to achieve stable performance (Wang et al., 2020; Guo et al., 2017). Here we show the base case, i.e. calculations with RH, temperature, major components and NH$_3$. Including gaseous HCl, gaseous HNO$_3$ and the effects of particulate organic nitrate did not substantially influence calculated AWC (see SI)." (Line 158–162)

Minor Comments

Line 71-79: References to online organic aerosol measurement using extractive electrospray ionization (EESI) technique (e.g. 10.5194/amt-12-4867-2019 and 10.5194/amt-14-1545-2021) should be added.

Reply: the references about EESI measurements were added

"Online organic aerosol measurements using the extractive electrospray ionization (EESI) technique could provide in-situ molecular composition (Lopez-Hilfiker et al., 2019; Pagonis et al., 2021)." (Line 76)

Line 105-108: A demonstration of IHNO3- time series would be helpful here.

Reply: the time series of HNO$_3$I$^-$ during the sampling period and the thermograms of Nov 8 were added to the revised supplementary information.

[Figure]

**Figure R9 (Figure S4)** (a)Time series of $HNO_3I^-$ integrated signals during the whole sampling period and (b) thermal desorption time series of the signals $HNO_3I^-$, $I^-$ and temperature for the sample of Nov 8.

Line 123-125: Because the aerosol composition was presented in rather semi-quantitative manners (i.e. percentage contribution by CHO vs. CHOX, or by individual elements), a note (even if somewhat qualitative) on potential sensitivity biases of I- CIMS should be added here. For instance, does I- CIMS respond equally well to organic acids, organosulfates, organonitrates, and reduced nitrogen species?

Reply: As the reviewer points out, the FIGAERO-CIMS sensitivity might depend on compounds' chemical composition. However, a general judgment on the sensitivity between organonitrate, organosulfate, and organic acids would be difficult since the sensitivity would vary among hundreds of individual compounds. The sensitivity of compounds depends on many factors such as ionization efficiency as well as transmission efficiency (Ye et al., 2021; Lopez-Hilfiker et al., 2016a). Whereas our results may be influenced by differences in sensitivity among individual compounds, we still observed good relative agreement between offline FIGAERO-CIMS organic compounds and ACSM total Org during the entire period of our measurements (Figure S3) despite changes in molecular composition. Considering the variations of sensitivity among different individual compounds, we revised the sentences in the main text as follows:

"Given this study's focus on the variability of the molecular composition of oxygenated OA and its relative changes, we did not attempt to convert total ion counts into atmospheric concentrations as the quantification of individual compounds is complicated by the variable sensitivities to different compounds (Lee et al., 2014)." (Line 129–131)

Line 141: How is NH3 detected by the NO3- CI-APi-TOF? Was there any consideration taken to minimize wall effects for the quantification of NH3, which can be quite "sticky"?

Reply: in our study, $NH_3$ was measured by a modified TOF-CIMS. Neutral $NH_3$ was charged by $H_3O^+$ or its hydrated clusters and calibrated by a commercial permeation device ($NH_3$ 1 min LOD: 0.08ppbv and $r^2$ of 0.997 between signals and $NH_3$ standards). A detailed description of $NH_3$ measurement can be found in Zheng et al. (2015). We corrected the description of $NH_3$ measurements in the main test as follows:

"Gaseous $NH_3$ was measured by a collocated modified Chemical Ionization–Atmospheric Pressure interface–Time Of Flight mass spectrometer (CI-APi-TOF, Aerodyne Research Inc., US) charged by $H_3O^+$ or its hydrated clusters. The $NH_3$ measurement method is described in previous studies (Cai et al., 2021; Zheng et al., 2015). " (Line 152–154)

Line 186: "P_BC > 23%". Should this be "P_BrC"?

Reply: Corrected. It now reads $P_{BrC}$.

Line 288: Does the TOF ACSM have enough mass resolving power for ion assignment? If not, the sentence should be revised to make the assumption here more explicit. Also, C2H4O+ would be found at m/z 44, not at m/z 60.

Reply: here it should be $C_2H_4O_2^+$ and we agree with the reviewer that the description of $f_{60}$ should not be very explicit due to the mass resolution of the TOF-ACSM. The sentence has been revised as follows to be more accurate:

"Another indicator for biomass burning, $f_{60}$, was measured by ACSM (Cubison et al., 2011)"

Line 315, 320, etc.: It would easier to write C6HxO5 (where x is a range of hydrogen atoms observed) instead of "C6 compounds with 5 oxygens"

Reply: the compounds are revised as suggested throughout the manuscript. (Line 330, 333, 334, 343)

Line 318: "C6HhO3Nn". Does "h" stand for anything in particular? If not, it would be clearer to write down the ranges of hydrogen atoms observed, e.g. C6H4-10O3Na

Reply: it has been revised to $C_6H_{5-11}O_3N$ as suggested. (Line 333)

Line 370-372: FIGAERO CIMS can only determine the elemental formula, not the molecular identity. The molecular identities (e.g. "malonic acid", "succinic acid", "glutaric acid") should be presented in less definitive tones.

Reply: Thanks for the suggestion. The sentence has been revised to: "other dicarboxylic acid-like compounds such as $C_3H_4O_4I^-$ (likely malonic acid), $C_4H_6O_4I^-$ (likely succinic acid), and $C_5H_8O_4I^-$ (likely glutaric acid) showed much higher (20–60 times) signals compared to the clean period." (Line 400–402)

Other descriptions related to molecular identities have been toned down as suggested throughout the manuscript.

Line 401-405: How consistent (with respect to OA loading and sources) is the POA vs SOA estimation based on ACSM measurements? How sensitive is the conclusion regarding POA vs SOA vs SIA effects to uncertainties in the f44 vs. f57 parameterization?

Reply: the POA and SOA separation in our study is based on the empirical equation proposed by Ng et al. (2011). From their study, it is found that the OA concentrations from the tracer-estimated study are within 30% of those from the positive matrix factorization (PMF)-AMS method for most sampling sites ($r^2$ =0.67 to 0.97). PMF analysis was not conducted in this study due to our relatively short sampling period (~2 weeks).

The conclusions on POA/SOA to OA optical effects in Beijing were already reported in previous studies (Zhang et al., 2018; Xie et al., 2019; Cheng et al., 2016). In this study, we followed earlier analysis procedures and results for bulk POA/OA components and optical parameters. Besides, our study further focused on the correlation between the OA compounds, which are identified by FIGAERO-CIMS, and particle optical parameters measured by AE-33. Thus, the uncertainties arising from POA/SOA separation based on ACSM data will not influence our main conclusions.